# Decision Support Algorithm Based on the Concentrations of Air Pollutants Visualization

**DOI:** 10.3390/s20205931

**Published:** 2020-10-20

**Authors:** Ekaterina Svertoka, Mihaela Bălănescu, George Suciu, Adrian Pasat, Alexandru Drosu

**Affiliations:** 1Department of Telecommunications, University Politehnica of Bucharest, 061071 Bucharest, Romania; esvertoka@elcom.pub.ro; 2Department of Telecommunications, Brno University of Technology, 61600 Brno, Czech Republic; 3Beia Consult International, 010158 Bucharest, Romania; mihaela.balanescu@beia.ro (M.B.); adrian.pasat@beia.ro (A.P.); alex.drosu@beia.ro (A.D.)

**Keywords:** Internet of Things, sensors, environment, monitoring, health, Air Quality Index, pollutant

## Abstract

As medical technologies are continuously evolving, consumer involvement in health is also increasing significantly. The integration of the Internet of Things (IoT) concept in the health domain may improve the quality of healthcare through the use of wearable sensors and the acquisition of vital and environmental parameters. Currently, there is significant progress in developing new approaches to provide medical care and maintain the safety of the life of the population remotely and around the clock. Despite the standards for emissions of harmful substances into the atmosphere established by the legislation of different countries, the level of pollutants in the air often exceeds the permissible limits, which is a danger not only for the population but also for the environment as a whole. To control the situation an Air Quality Index (AQI) was introduced. For today, many works discuss AQI, however, most of them are aimed rather at studying the methodologies for calculating the index and comparing air quality in certain regions of different countries, rather than creating a system that will not only calculate the index in real-time but also make it publicly available and understandable to the population. Therefore we would like to present a decision support algorithm for a solution called “Environmental Sensing to Act for a Better Quality of Life: Smart Health” with the primary goal of ensuring the transformation of raw environmental data collected by special sensors (data which typically require scientific interpretation) into a form that can be easily understood by the average user; this is achieved through the proposed algorithm. The obtained result is a system that increases the self-awareness and self-adaptability of people in environmental monitoring by offering easy to read and understand suggestions. The algorithm considers three types of parameters (concentration of PM10 (particulate matter), PM2.5, and NO_2_) and four risk levels for each of them. The technical implementation is presented in a step-like procedure and includes all the details (such as calculating the Air Quality Index—AQI, for each parameter). The results are presented in a front-end where the average user can observe the results of the measurements and the suggestions for decision support. This paper presents a supporting decision algorithm, highlights the basic concept that was used in the development process, and discusses the result of the implementation of the proposed solution.

## 1. Introduction

Today’s society generates an increased amount of air pollutants from multiple sources: industrial waste, transport, dust, household energy, etc. Despite the framework restricting the emission of pollutants into the atmosphere established by law, a significant amount of mixtures enters the air every day. Numerous studies confirm the direct relationship between air pollution and diseases such as lung cancer, stroke, pneumonia, ischemic heart disease, etc. According to the World Health Organization (WHO), 7 million people die prematurely annually from air pollution [1]. Therefore, the search for new solutions to reduce this number, while a complex process, is an urgent task.

In this work, we aim to advance a system of improving quality of life through an IoT computer system that collects and analyzes data from networks of environmental sensors, wearable (portable) sensors, IT applications, and the human factor [2] and represent the results in an understandable for everyone manner [3].

Thus, the main objective is to ensure the transformation of raw environmental data collected by special sensors and requiring the interpretation of a specialist into a form that can be easily understood by an ordinary user. The raw environmental data here mean the measurements in particular areas of three pollutant concentrations—PM10, PM2.5, and NO_2_—based on which the Air Quality Index (AQI) is calculated. AQI shows a quantitative level of air pollution in a studied area. This index could be interpreted in the form of a color scale of four colors (green—low level of air pollution; yellow—moderate; orange—high; red—very high) and text recommendations for users of two classes: the general population and people at risk. The proposed solution was tested in the pilot area in Bucharest, Romania.

While the concepts of collecting and processing air quality data are not new, this particular approach is designed to take into account an increased number of data acquisition devices, e.g., different types of wearable devices or surface mounted sensors, etc. It also transforms the information into clear and concise suggestions and statements so that different types of users, e.g., elderly, adults, etc., could acknowledge their location status and take corresponding measures. For this, the proposed algorithm considers parameters such as PM10, PM2.5, NO_2_, and the calculated AQI. Based on their concentration, the mentioned pollutants are divided into four classes (from low to very high), further modifying the scientific area approach into a more familiar and user-friendly language. 

Air pollution affects people in different ways due to numerous factors, such as age, lung capacity, health status, and time spent in the environment, which are taken into account. Moreover, different impurities in the air can also have different effects; for example, large particles of pollutants can be filtered out by the human body, while small particles penetrate the lung alveoli [4]. 

The proposed solution will allow its users to adapt faster and easier to the environment, depending on one’s health and physical needs. By using the developed platform, the user will obtain access to individual recovery programs, developed based on the assimilation of data integrated into the platform from environmental and portable sensors [5]. Thus, the creation of a user profile that takes into account the psychological state, preferences, and behavior (self-awareness and self-adaptations) is aimed at shortening the recovery period, as well as allowing for a seamless integration into society and the work environment.

This paper is organized as follows. Section 2 considers recent related research in the field of study. Section 3 highlights the basic concepts and approaches utilized during the system’s development. Section 4 presents the developed architecture. Section 5 describes the proposed decision support algorithm and discusses the results of its implementation. Section 6 presents a final discussion and conclusions.

## 2. Related Work

Healthcare is one of the areas where technology is evolving very rapidly due to necessity and sensors, just like the wearable devices that make up the Internet of Health Things (IoHT), are no exception. This field is increasing faster than all the others due to the increasing life level, level of medicine, and money savings because of proper treatment [6]. At the moment, there are no areas of healthcare left where wearable technology would not be involved, from conventional heart rate monitors to sophisticated systems for tracking severe cases, such as e.g., patients with Parkinson’s [7] or Alzheimer’s [8] diseases, or epileptic seizures [9].

The use of medical data is the IoT concept is also addressed in [10] where the authors propose an innovative framework meant to protect this type of highly sensitive and personal data. As the volume of patients, procedures are health parameters are continuously increasing so is the need for security towards it. The proposed framework is focused on ECG data.

Although the proposed solution is not directly related to healthcare, it attempts to improve the general level of health of the population, especially helping those who suffer from lung or heart diseases. In addition, the assessment of air pollution is very important in the framework of industries, especially hazardous ones, where emissions very often can exceed an acceptable level. Knowing this, safety assessment managers can timely evacuate workers and provide proper treatment. The Environmental Protection Agency (EPA) introduced the concept of the Pollutant Standard Index (PSI) for the assessment of the air quality in 1976. PSI has not been standardized and different areas of America and Canada used different indexes, which led to confusion. In 1999, EPA unified the index and renamed it from PSI to AQI.

AQI is the highest value for five measures pollutants: O_3_ (ozone), PM, CO (carbon monoxide), SO_2_ (sulfur dioxide), and NO_2_ (nitrogen dioxide). This index varies from 0 to 500 and six classes (and six colors) are distinguished and depend on the value: good (0–50, green), moderate (51–100, yellow), unhealthy for sensitive groups (101–150, orange), unhealthy (151–200, red), very unhealthy (201–300, purple), and hazardous (301–500, maroon) [11]. 

Initially, this index had to be reported by a state if a metropolitan statistical area (MSA) had more than 350,000 people. As of 2013, about 20 countries have started to use AQI to assess the degree of pollution in the air [12]. Currently, the number of countries tracking this parameter exceeds 50 [13]. However, the AQI calculated in one country is not necessarily equal to AQI calculated in another because the list of monitored pollutants, the way they are measured, the air quality guidelines, and the methodology for calculating the index itself can be different. The methods of making measurements will be discussed in more detail in Section 3.

Generally, when talking about the air quality, it is essential to specify two different cases: indoor and outdoor. In an indoor environment, the air state strongly depends on the quality of the ventilation system. Studies [14,15,16,17] investigated the level of air pollution in underground conditions in China, Italy, Romania, and Greece consequently. All works consider the assessment of the air condition based on a level of PM, the pollutant with the highest exceeding over WHO’s safe limits [16]. Most of the studies report that concentrations of PMs measured in subway stations are much higher than the same concentrations measured outdoor due to mechanical processes in trains. In contrast, this work is mainly focused on the estimation of the air condition in the outdoor environment. Some of the main sources of air pollution here are industrial emissions, power plants, and exhaust gases. This case also includes the estimation of air pollution level via AQI; however, the areas are more significant, the list of tracked pollutants is wider, and measured results are usually less severe than in the closed space. 

The number of works that investigate AQI in the outdoor environment in different countries is vast, differing in the applied methodology of calculating the index, the way of interpretation, and the number of pollutants measured in the atmosphere.

For example, the authors of [14] propose a modified AQI interpretation method for Tunisia. They measure four pollutants (SO_2_, NO_2_, PM_10_, and O_3_) during a particular time, take the maximum value to identify AQI, and then, rate it on a five-point scale (bad, poor, acceptable, good, and very good). This scale is defined by four thresholds: air quality alarm value, values which exceed Tunisian limiting values, Tunisian limiting value, and guide values recommended by the WHO. This study does not distinguish sensitive groups.

The work [18] considers five methods of AQI calculation: as an arithmetic mean of the ratio of pollutant measured in the air to its standard value (with and without weights for pollutants), as a geometric mean of the same ratio, according to the Oak Ridge National Laboratory formula, and according to the algorithm proposed by EPA. All methods were tested in India. The study does not conclude which method is more appropriate, but identifies the most dangerous pollutant, namely PM, based on the results.

The authors of [19] estimate AQI in several cities of India (Chennai, Bangalore, and Delhi), using hourly measurements of PM, SO_2_, NO_2_, and RSPM (Respirable Suspended PM). In this work, the AQI is calculated in two ways: in the first case, the index is the maximum value of all pollutants, and in the second, the index is the average value. The hazard classes are distributed in the same way as suggested by EPA in 1999.

To the best of the authors knowledge, most of the works devoted to the assessment of AQI are aimed at studying the methodologies for calculating the index and comparing air quality in certain regions of different countries, rather than creating a system that will not only calculate the index in real-time but also make it publicly available and understandable to the population. The proposed solution allows the combination of both functions. The next sections reveal the basic concepts, method of AQI calculation, and interpretation used in the developed system and the results of its implementation.

## 3. Underlaying Concepts

### 3.1. Internet of Things (IoT)

IoT represents a system consisting of linked computing devices, mechanical and digital machines, objects, and living creatures that have a unique identifier (UID) and can transmit data through a network in the absence of human-to-human or human-to-computer contacts [6,20].

The concept of IoT is growing very fast, especially in the following areas: healthcare, transport, industrial automation, and smart houses. The healthcare domain relies on the IoT through its special niche—the Internet of Health Things (IoHT) [6]. This field is developing faster than all the others due to increasing quality of life, level of medicine, and money savings because of timely informing and better treatment. 

Generally, the IoHT provides a connection between patient and healthcare facilities.

### 3.2. Smart Health

Smart Health is a concept that involves the use of high-performance devices to improve medical treatments and, consequently, quality of life. Smart Health enhances the patient–medic relationship and empowers patients through actionable insights [21].

Some examples of Smart Health platforms are: Kaa IoT is a versatile, multifunctional, open-source platform for the implementation of complete IoT solutions, connected applications, and intelligent products. Kaa has professional-grade IoT functions that can be connected and used to implement a vast majority of IoT use cases [18].Vista data vision is a comprehensive software solution perfect for data management and visualization environmental monitoring projects. Data are sent from the data logger to a server, where they are automatically converted into a system-compatible format and imported into the database. Then, the data can be viewed on the web-based interface [22].InteliS integrates the most critical devices and machines in order to find solutions for remote diagnostics, preventive maintenance, resource optimization, quality improvement, or delivery time reduction [23].

There are several reasons to use IoT in Smart Health, such as easy data access, flexible communications between those who use the platform (e.g., patient–medic), and autonomous actions that significantly save such a scarce resource as time.

The general principle of the IoT for health utilization is as follows: patients wear sensors, which collect different types of data related to health such as heart rate, electrocardiography, etc. Data are then processed by applications of the User Terminals (UT). UTs may be connected with the gateway via short-range communication protocols, e.g., Bluetooth Low Energy [24,25]. The gateway is connected with the cloud or medical service, where the collected data are stored or processed. Data can also be located in Electronic Health Records (EHR), from which they could easily exported by, e.g., a doctor during the patient’s visit. 

To summarize, the key benefits of using IoT in Smart Health are the following:Remote supervision. Gadgets can measure the patient’s vital parameters and the corresponding data could be associated with the user’s profile, where nurses and doctors can access them, analyze, and give feedback.Personal care [23]. Wearable sensors can be used to track the individual’s changes in health state. With these sensors, people with chronic diseases such as diabetes can monitor whether they are keeping their condition under control or not [26].Building smart hospitals [27]. In such institutions, doctors use their smartphones as assistants to manage the data. The data do not require to be entered manually since the devices automatically collect and store them. In addition, the waiting times for patients are reduced [28].

### 3.3. Air Quality Index

Air quality is measured with the AQI. This parameter illustrates changes in the amount of pollution in the atmosphere. In general, we calculate AQI for major air pollutants: particle pollution, ground-level ozone, carbon monoxide, nitrogen dioxide, and sulfur dioxide. We use AQI to receive information about air quality and, consequently, to find out how to protect personal health.

There are different ways to measure AQI, for example, with: Portable particulate monitor PM 10/PM 2.5 [29];Huma-i HI-150 (advanced portable air quality monitor indoor/outdoor that measures CO_2_, volatile organic compounds (VOC), particle matter, temperature, and humidity);“Air quality meter” application that can measure the PM10 in an outdoor environment using a smartphone camera and many others.

One of the examples of an AQI estimation algorithm is PM NowCast. It was designed by the United States Environmental Protection Agency (USEPA) and is used for real-time reporting of the Air Quality Index (AQI) for PM (PM10 or PM2.5) [30,31,32,33]. In a general case, PM NowCast is calculated as

(1)
$$Nowcast=\frac{\sum_{i=1}^{12}w^{i-1}c_{i}}{\sum_{i=1}^{12}w^{i-1}}$$

where 
$c_{1},\;c_{2}\;\ldots \;c_{12}$
 are PM concentrations over the last 12 h consequently (
$c_{1}$
 is the most recent value), and 
w
 is a variable that is calculated as

(2)
$$w=\left\{\begin{array}{c} w^{*}\;\;if\;1-w^{*}>\frac{1}{2}\;,\;\\ \frac{1}{2}\;\;if\;1-w^{*}\leq \frac{1}{2}\;. \end{array}\right\},$$


(3)
$$w^{*}=\frac{c_{max}-c_{min}}{c_{max}}\;,$$

where 
$c_{max},\;c_{min}$
 are the maximum and the minimum PM concentration over the last 12 h, consequently. 

Based on the calculated PM NowCast, it is possible to determine AQI as

(4)
$$AQI=\frac{Nowcast-C_{low}}{C_{high}-C_{low}}\left(I_{high}-I_{low}\right)-I_{low},\;$$

where 
$C_{low}$
 is the concentration breakpoint that is 
≤ Nowcast
, 
$C_{high}$
 is the concentration breakpoint that is 
> Nowcast
, 
$I_{low},\;I_{high}$
 are the indexes breakpoints corresponding to 
$C_{low},\;C_{high}$
 consequently. Based on the values of these parameters, a color code is defined (see Table 1). A color code is a straightforward and understandable indicator that assists people with no scientific background with evaluating the air quality in a certain location and take measures if it poses a danger to them.

The interpretation of AQI for users in Table 1 involves dividing the population into two groups: the sensitive and the non-sensitive. Air pollution affects people differently and in the best case, a number of different factors such as age, lung diseases, lung capacity, etc., should be taken into account. In the proposed solution, one group is distinguished among the entire population, called “sensitive”, which includes people with heart and/or lung diseases, asthma, circulatory disorders, elderly people, children, and athletes whose performance may decline due to insufficient air quality.

## 4. System Architecture

The platform contains the following hardware and software components (see Figure 1):A module for the collection of data from the environment equipped with several sensors that allow the monitoring of the following parameters: relative humidity, air temperature, atmospheric pressure, suspended dust concentrations, and concentrations of gaseous pollutants (SO_2_, NO_2_, CO, CO_2_, VOC).A multiprotocol gateway module that allows for collection of the data from the sensors and ensures their transmission to the cloud through Ethernet/4G/3G/GPRS communication protocols [33].An intermediate data transmission component that operates as the communication medium between the gateway and the data persistence level possible.The decision support component [34], which is a module for processing the data collected from the sensors. The result of the decision support component analysis is transmitted further to the data presentation and visualization mode.A data visualization module: this module is a UT, where the information can be observed [35]. The data displayed include the values of monitored pollutants, contextual messages for the general and sensitive population, and a chart with the values registered in the last hours.

The proposed solution supports the following categories of sensors:Sensors and instruments for measuring chemical compounds in the atmosphere and weather sensors [36]. The developed web application queries these sensors through a set of Application Programming Interfaces (APIs) that model a Representational State Transfer (REST) mechanism. Although some models are portable, due to the need to report a measurement in a particular area, the environment sensors are defined as a fixed type. They have an associated area in the system that can be an indoor location or a surface on a map.Garmin IoT Wearable bracelets are utilized for measuring the factors related to the physical and health status of the wearer [37]. These sensors are defined as mobile and are represented by the combination of a bracelet and a smartphone. The bracelet monitors the vital parameters of a user and transmits these data via BLE to the Garmin app installed on the smartphone. Through an API, the platform loads the data provided by the Garmin mobile application.Smartphones for detecting and reporting the presence of a user in an outdoor area using Global Positioning System (GPS) technology or in an indoor area using BLE beacons [38,39]. These sensors are also defined in the platform as mobile. In this case, the sensor is the smartphone that sends information about the presence, proximity, and location on the platform. The user needs to download the mobile application and activate it. Next, the user can go with it to various locations. The platform uploads the data provided by the mobile app through an API.In addition to the values measured by sensors, the developed platform will be also able to use data from an external source (through specialized interfaces) such as files exported from other platforms and applications.

Other wearable sensors are used to enable long-term continuous physiological and environmental monitoring, which is vital for the treatment and management of many chronic illnesses, neurological disorders, and mental health issues. In addition to providing a solution for real-time classification outside the lab, wearable sensors can provide an efficient way to collect a large amount of data. 

## 5. Developed Decision Support Component

This section outlines the decision support algorithm description and provides some implementation details as well as some numerical results.

### 5.1. Algorithm Description

The main tasks of decision support are measuring AQI and its subsequent transformation into an easily interpretable form for any user. In the proposed algorithm, the composite AQI will be evaluated, taking into account the concentration values of all atmospheric pollutants. Then, it will be transformed into a color scale, with which everyone can understand the results and assess the risk of being in a particular area regarding air quality. 

When converting the index to a color scale, two components are distinguished:The main element that is responsible for the color visualization. It is provided by the pollutant with the highest concentration compared to the limit values;The secondary component is responsible for the color intensity. This component is determined by the levels of the other pollutants.

The proposed decision support algorithm has four main operational steps. The flowchart of the algorithm is given in Figure 2, while the steps are detailed below.

Step 1: Assigning a risk class 
$K_{1}$
…
$K_{i}$
 to each pollutant 
$P_{1}$
… 
$P_{n}$
 based on its average concentration measured hourly 
${\bar{v}}_{n}$
 (see Table 1). There are four classes of risk severity of the negative impact of the concentration of pollutants on human health: low, moderate, high, and very high. Each of them corresponds to a predetermined limit concentration for each pollutant 
$v_{1\ldots 4,n}$
, and each of them is assigned a specific color—green, yellow, orange, and red—in accordance with the severity of the risk. Note, the limit concentration value for the fourth class 
$v_{4,n}$
 is not defined and such a record is used for presentation ease. 

Based on the average concentration measured during an hour 
${\bar{v}}_{n}$
 in Table 2, each pollutant is assigned a specific class; see example in Table 3.

Step 2: Determining the maximum risk class number 
$i_{max}$
, 
$i\in \left[1\ldots 4\right]$
. For this, class, the values are sorted in descending order, after which the maximum value is selected. For instance, in the example presented in Table 3, 
$i_{max}$
 is 4.

Step 3: Calculation of the AQI composite index 
$AQI_{comp}$


(5)
$$\begin{array}{cc} AQI_{comp} & =\underbrace{i_{max}}+\underbrace{\left(\frac{\sum_{j=1}^{n}i_{j}-i_{max}}{n-1}\right)/i_{max}}\\ & I_{color}\;I_{intensity} \end{array}$$


Step 4: Determining the basic color and intensity of the 
$AQI_{comp}$
. As could be observed from (5), the basic color index 
$I_{color}$
 of 
$AQI_{comp}$
 is given by the color associated with the maximum value class—
$K_{max}$
. Its intensity is determined according to the value of the intensity index 
$I_{intensity}$
. Four gradations of intensity are provided as

(6)
$$intensity=\left\{\begin{array}{cc} 25\;\% & 0<\;I_{intensity}<0.25,\;\\ 50\;\% & 0.25\leq \;I_{intensity}<0.50,\\ 75\;\% & 0.50\leq \;I_{intensity}<0.75,\\ 100\;\% & 0.75\leq I_{intensity}<1.00. \end{array}\right.$$


### 5.2. Decision Support Component Implementation

The decision support component uses a complex structure of hardware and software subcomponents [40]. The hardware subcomponents are backend infrastructure, environmental sensors, and gateway equipment (Libelium Meshlium) [41]. Software components are represented by the LAMP package (Linux, Apache, MySQL, PHP) [42].

For the implementation of the decision support component, these steps were followed (see GitHub Repository for details: https://gist.github.com/BotezatuAndreea):
Extracting data from GM Meshlium’s MySQL database [29]:
The query for connection verification with the GW database;The query for retrieving values (PM10, PM2.5, and NO2 parameters) [30] and calculating the moving average.Application of the air quality index calculation algorithm.Storing the values calculated above.Displaying the AQI graph for the last 24 h and the decision support component.

### 5.3. Experimental Results

The proposed decision support algorithm was implemented and tested in Bucharest, Romania. The tested pilot area is depicted in Figure 3.

In order to transfer the calculated AQI to the color scale, the tables of concentration limit values from [43] were used (see Table 4).

After concentration measuring pollutants are distributed by the class, 
$K_{max}$
 and AQI are determined. Based on the results, the recommendations both for people in danger and the general population are provided in Table 5 [43].

The results of the implementation are given in Figure 4. In the current version, the system measures concentrations of three pollutants (PM10, PM2.5, NO_2_), calculates AQI, transfers it to the color scale that takes into account just the color index 
$I_{color}$
, and displays the results in the form of a bar chart, where the index value is plotted along the ordinate axis, and time is plotted along the abscissa axis (the discrete of the abscissa axis is one hour). In addition to the bar chart, the visual component of the decision support system, which will be displayed on the UT, includes the values of monitored pollutants based on which the AQI was calculated (PM10, PM2.5, NO_2_) on the bottom right and contextual messages for the general and sensitive population from Table 4 on the bottom left.

In terms of limitations, it is significant to mention that although consistent results were obtained using this approach, only a limited amount of data and data acquisition devices were considered. Even if the system was designed and focused on scalability (by using different APIs corresponding to data acquisition devices and a REST approach), further testing (previsioned as future work) is required in order to confirm the compiling with the big data type of information flow. Thus, despite what was proposed in Section 5.1, the algorithm is not yet finalized, but the pilot system successfully achieves the main goal, namely, improving quality of life by collecting and analyzing data from networks of environmental sensors, wearable (portable) sensors, IT applications, and the human factor, while also representing the results in an understandable for everyone manner.

## 6. Discussion

This paper proposes a decision support algorithm for air pollution monitoring visualization. The development of this component involved finding and taking into account the relationship between the concentrations of pollutants in the atmosphere and health effects, quantification methods, and notifications for sensitive people and the general population. Its target envisages calculating the AQI and its subsequent transformation into a color scale that can be easily understood by ordinary users. 

The proposed solution allows a quick response and avoidance of places where air quality reaches levels that can harm people of different groups. The population is provided with easy-to-interpret notifications of the current state of air pollution on a graphical interface. Hospital departments specialized in lung problems using this AQI calculator also can operatively react and warn their patients about the potential danger. Thus, the proposed solution is expected to increase the general level of public health. This system cannot reduce the number of pollutant emissions into the atmosphere directly, but by providing information about the current state of the air in real-time in a convenient form publicly, it can push society to create the necessary conditions for changing the situation at the legislative level.

In the future, we plan to implement the entire algorithm including the intensity variety that was mentioned in Section 4, step 4. This modification will increase the degree of scale gradation, allowing one to determine the AQI more accurately. The extended scale of classes will also require a more detailed investigation of the degree of influence of the studied pollutants on people with different health states in order to make more thorough recommendations. The final step will be testing and optimization of the system in the same location.

## Figures and Tables

**Figure 1 sensors-20-05931-f001:**
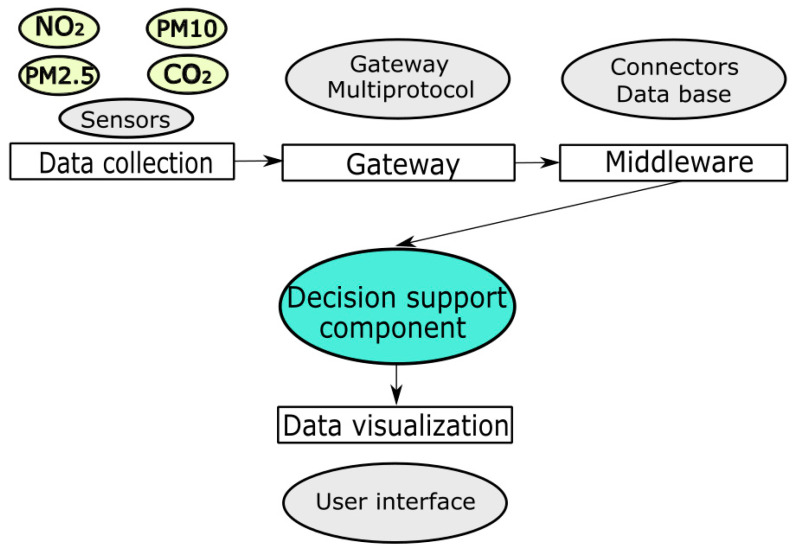
The architecture of the ESTABLISH (Environmental Sensing To Act for a Better Quality of Life: Smart Health) solution.

**Figure 2 sensors-20-05931-f002:**
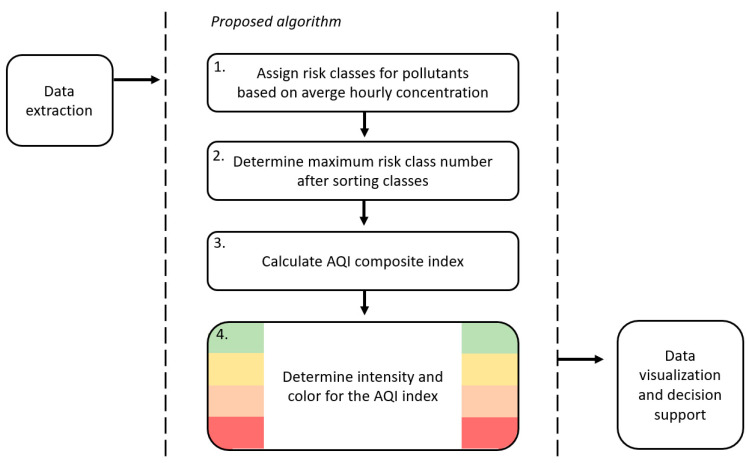
Flowchart for the decision support algorithm.

**Figure 3 sensors-20-05931-f003:**
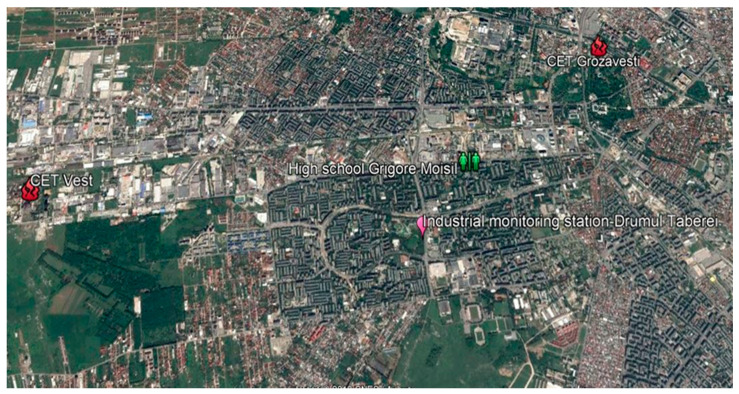
Developed solution pilot area.

**Figure 4 sensors-20-05931-f004:**
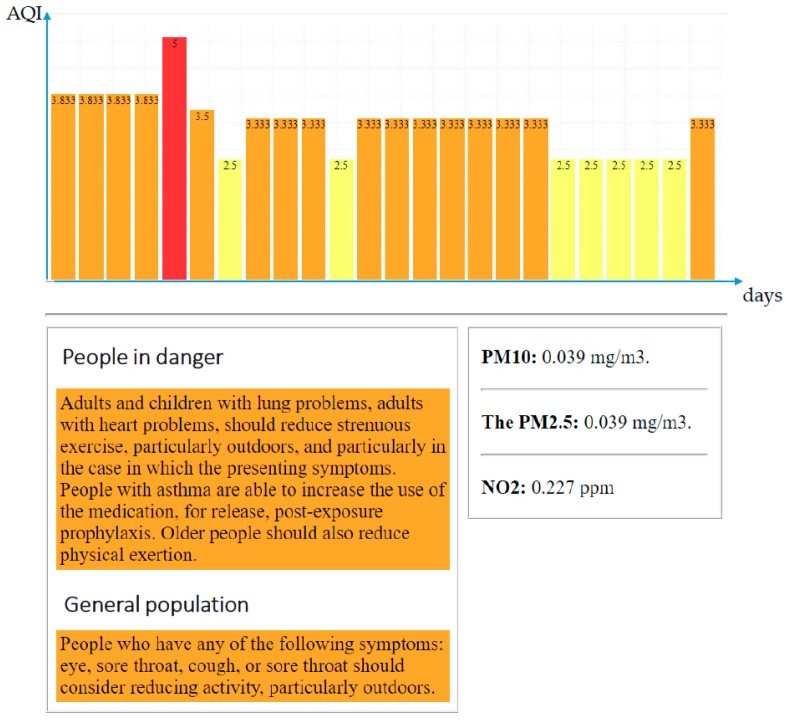
Real-time representation of decision support component for two cases.

**Table 1 sensors-20-05931-t001:** Categories and indicators for AQI [16].

*C_low_*	*C_high_*	*I_low_*	*I_high_*	Category
0	12.0	0	50	Good
12.1	35.4	51	100	Moderate
35.5	55.4	101	150	Unhealthy for Sensitive Groups
55.5	150.4	151	200	Unhealthy
150.5	250.4	201	300	Very Unhealthy
250.5	350.4	301	400	Hazardous
350.5	500.4	401	500	Hazardous

**Table 2 sensors-20-05931-t002:** Model table of concentration limits for each class.

Class	Pollutant Concentration (µg/m^3^)			
	*P*_1_	*P*_2_	*…*	*P_n_*
$K_{1}$ (low)	$[0\ldots v_{11}$ ]	$[0\ldots v_{12}$ ]	*…*	$[0\ldots v_{1n}$ ]
$K_{2}$ (moderate)	$(v_{11}$ *…* $v_{21}$ ]	$(v_{12}$ *…* $v_{22}$ ]	*…*	$(v_{1n}$ *…* $v_{2n}$ ]
$K_{3}$ (high)	$(v_{21}$ *…* $v_{31}$ ]	$(v_{22}$ *…* $v_{32}$ ]	*…*	$(v_{2n}$ *…* $v_{3n}$ ]
$K_{4}$ (very high)	$(v_{31}$ *…* $v_{41}$ ]	$(v_{32}$ *…* $v_{42}$ ]	*…*	$(v_{3n}$ *…* $v_{4n}$ ]

**Table 3 sensors-20-05931-t003:** Distribution of pollutants by class.

Class	Pollutant			
	*P*_1_	*P*_2_	*…*	*P_n_*
$K_{1}$ (low)	✓		…	
$K_{2}$ (moderate)		✓	…	
$K_{3}$ (high)			…	
$K_{4}$ (very high)			…	✓

**Table 4 sensors-20-05931-t004:** Table of concentration limits for each class used in the ESTABLISH solution [43].

Class	Pollutant Concentration (µg/m^3^)		
	PM10	PM2.5	NO_2_
$K_{1}$ (low)	[0…50 ]	[0…35 ]	[0…200 ]
$K_{2}$ (moderate)	(50…75 ]	(35…53 ]	(200…400 ]
$K_{3}$ (high)	(75…100 ]	(53…70 ]	(400…600 ]
$K_{4}$ (very high)	(100…∞ )	(70…∞ ]	(600…∞ ]

**Table 5 sensors-20-05931-t005:** Recommendations [43].

Class	Recommendations	
	People in Danger	General Population
$K_{1}$ (low)	Enjoy your usual outdoor activities.	Enjoy your usual outdoor activities.
$K_{2}$ (moderate)	Adults and children with lung problems and adults with heart problems, who experience symptoms, should consider reducing strenuous physical activity, particularly outdoors.	Enjoy your usual outdoor activities.
$K_{3}$ (high)	Adults and children with lung problems and adults with heart problems should reduce strenuous physical exertion, particularly outdoors, and particularly if they experience symptoms. People with asthma may find they need to use their reliever inhaler more often. Older people should also reduce physical exertion.	Anyone experiencing discomforts such as sore eyes, cough, or sore throat should consider reducing activity, particularly outdoors.
$K_{4}$ (very high)	Adults and children with lung problems, adults with heart problems, and older people should avoid strenuous physical activity. People with asthma may find they need to use their reliever inhaler more often.	Reduce physical exertion, particularly outdoors, especially if you experience symptoms such as cough or sore throat.

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
