# Peer review of "Decision Support Algorithm Based on the Concentrations of Air Pollutants Visualization"

_sensors, 2020, doi:10.3390/s20205931_

Round 1

Reviewer 1 Report

This article presents a decision 12 support algorithm for a project called “Environmental Sensing To Act for a Better Quality of Life: 13 Smart Health” (ESTABLISH). The main objective of the project is to ensure the transformation of raw environmental data collected by special sensors and requiring the interpretation of a specialist  into a form that can be easily understood by an ordinary user.

The paper is well written and technically sound, however following comments need to be address before recommending it for publication.

1 - the abstract part need to included some quanitative results

2 - The introduction part is too vague, it does not reflect the whole idea and background of the article.

3- a separate section indicating state of the art comparison should be included

4- Reference part need to be revised as well, for example following state of the art work is missing.

Shah, S.A., Fan, D., Ren, A. et al. Seizure episodes detection via smart medical sensing system. J Ambient Intell Human Comput (2018).

Author Response

Dear reviewer,

Thanks for your comments!

Point 1: The abstract part need to included some quantative results

Response 1:  Have not been changed. The point is that we do not have quantitative results: we have one pilot area in Romania where the data is constantly measured. Based on the measurements we have a new AQI hourly. This system transfers AQI to a particular classes of danger each of which has its own color and accompanying recommendation.

Point 2: The introduction part is too vague, it does not reflect the whole idea and background of the article.

Response 2 (page (P) 1, line (L) 27-37): The introduction was extended to specify in more detail the sense of the proposed solution and how it works.

Point 3: A separate section indicating state of the art comparison should be included

Point 4: Reference part need to be revised as well, for example following state of the art work is missing. Shah, S.A., Fan, D., Ren, A. et al. Seizure episodes detection via smart medical sensing system. J Ambient Intell Human Comput (2018).

Response 3-4 (P 2): An additional section (Section 2) named “Related work” was presented. The list of the references was revised in accordance.

Reviewer 2 Report

Authors have highlighted the emerging and core issue, but still there are major issues to be fixed.

Reviews to Authors

  • Title must be simple, clearer and nicer.
  • Spell out each acronym the first time used in the body of the paper. Spell out acronyms in the Abstract by extending it.
  • The abstract can be rewritten to be more meaningful. The authors should add more details about their final results in the abstract. Abstract should clarify what is exactly proposed (the technical contribution) and how the proposed approach is validated.
  • What is the motivation of the proposed work?
  • Introduction needs to explain the main contributions of the work clearer.
  • The novelty of this paper is not clear. The difference between present work and previous Works should be highlighted.
  • Authors must explain in detail the introduction section.
  • Authors must develop the framework/architecture of the proposed methods
  • There is need of flowchart and pseudocode of the proposed techniques
  • Proposed methods should be compared with the state-of-the-art existing techniques
  • Research gaps, objectives of the proposed work should be clearly justified.
  • The authors should consider more recent research done in the field of their study (especially in the years 2017 and 2018/2019 onwards). To improve the introduction and related work authors are highly recommended to put the latest bibliography < A Multi-sensor Data Fusion Enabled Ensemble Approach for Medical Data from Body Sensor Networks’, Information Fusion, Elsevier, 53, No.2020, pp.155-164, 2020>, <An Energy-Efficient Algorithm for Wearable Electrocardiogram Signal Processing in Ubiquitous  Healthcare Applications”, MDPI Sensors Vol.8, No.3, pp.923, 2018>
  • English must be revised throughout the manuscript.
  • Limitations and Highlights of the proposed methods must be addressed properly
  • Experimental results are not convincing, so authors must give more results to justify their proposal.

Finally, paper needs major improvements

Author Response

Dear reviewer,

Thanks for your comments!

Point 1: Title must be simple, clearer and nicer.

Response 1: The title was changed to the simpler and more understandable one.

Point 2: Spell out each acronym the first time used in the body of the paper. Spell out acronyms in the Abstract by extending it.

Response 2: Solved, all the acronyms in the paper are first spelled out.

Point 3: The abstract can be rewritten to be more meaningful. The authors should add more details about their final results in the abstract. Abstract should clarify what is exactly proposed (the technical contribution) and how the proposed approach is validated.

Response 3 (page (P) 1, lines (L) 28-33): Abstract was modified accordingly.

Point 4: What is the motivation of the proposed work?
Response 4 (P 2, L 42-51): Motivation was further detailed in the Introduction section.

Point 5-7: Introduction needs to explain the main contributions of the work clearer. The novelty of this paper is not clear. The difference between present work and previous Works should be highlighted. Authors must explain in detail the introduction section.

Response 5-7 (P 2, L 56-72): Contributions were additionally explained. An additional part was added to Introduction section to reveal the novelty of the work.

Point 8: Authors must develop the framework/architecture of the proposed methods
Response 8: Architecture is available in Figure 2.

Point 9: There is need of flowchart and pseudocode of the proposed techniques

Response 9 (P 8, L 300): Added Figure 3 which presented the named flowchart.

Point 10: Proposed methods should be compared with the state-of-the-art existing techniques

Point 11: Research gaps, objectives of the proposed work should be clearly justified.

Point 12: The authors should consider more recent research done in the field of their study (especially in the years 2017 and 2018/2019 onwards). To improve the introduction and related work authors are highly recommended to put the latest bibliography < A Multi-sensor Data Fusion Enabled Ensemble Approach for Medical Data from Body Sensor Networks’, Information Fusion, Elsevier, 53, No.2020, pp.155-164, 2020>, <An Energy-Efficient Algorithm for Wearable Electrocardiogram Signal Processing in Ubiquitous  Healthcare Applications”, MDPI Sensors Vol.8, No.3, pp.923, 2018>

Response 10-12 (P 2): An additional section “Related work” was added. This section reviews the modern works related to the assessment of air quality. The proposed works were mentioned as a part of this section.

Point 13: English must be revised throughout the manuscript.

Response 13: The whole paper was fully revised.

Point 14: Limitations and Highlights of the proposed methods must be addressed properly

Response 14 (P 13, L 370-374): Detailed the issues in Section 5 and 6.

Reviewer 3 Report

Good, work, Please find attached my pdf file with some comments/suggestions than may help to improve the paper.

Author Response

Point 1 (page (P) 1, line (L) 8-9): Has not been changed: the text is just highlighted, but not commented.

Point 2 (P1, L11-15): Has not been changed: the text is just highlighted, but not commented.

Point 3 (P1, L20): Has not been changed: the text is just highlighted, but not commented.

Point 4 (P1, L30-32): Has not been changed (Thank you for the comment!).

Point 5 (P2, L108): Reference 15 was deleted, the reference list was updated.

Point 6 (P2, L112): The reference on the official source with NowCast calculation was added, the reference list was updated.

Point 7 (P7, L227): Has not been changed (Thank you for the comment!).

Point 8 (P7, L252-253): Axes were added, the quality of the picture was improved. The explanation is quite a detail, in my point of view (I do not know what can be added here), however, it was extended a little bit to explain the parts of the dashboard.

Point 9 (P1, L268-270): Has not been changed: the text is just highlighted, but not commented.

Point 10 (P1, L272-273): The more detailed information about the benefits of using the proposed solution.

Point 11 (P1, L277-278): The paragraph with the plans was expanded with a more detailed description of what exactly needs to be done.

Round 2

Reviewer 1 Report

The authors have addressed all my comments, I would recommend it for publication in present form.

Author Response

Dear reviewer!

Thank you for taking the time to help us improve our article for publication!

Best Regards,

Ekaterina Svertoka

Reviewer 2 Report

Authors have improved the paper, but there are still major changes to be fixed 

  1. Experimental results are not sufficient and convincing to justify their proposed method
  2. Pseudocode of flowchart of the proposed technique are not presented, so authors are highly recommended to add these for the better and high scope of the topic
  3. Intro and related sections are still weak, so they are suggested to improve these parts with latest and high quality works as, <A Multi-sensor Data Fusion Enabled Ensemble Approach for Medical Data from Body Sensor Networks>, <Towards Machine Learning Enabled Security Framework for IoT-based Healthcare>
  4. Introduction needs to explain the main contributions of the work clearer.
  5. The novelty of this paper is not clear. The difference between present work and previous Works should be highlighted.
  6. Authors must explain in the introduction section.
  7. Authors must develop the framework/architecture of the proposed methods
  8. There is need of flowchart and pseudocode of the proposed techniques
  9. Proposed methods should be compared with the state-of-the-art existing techniques

Major changes are recommended 

Author Response

Dear Reviewer,

Thank you for the comments!
You can find our point-to-point response below in the attached file!

Best Regards,

Ekaterina Svertoka

Round 3

Reviewer 2 Report

Paper has been significantly improved, I recommend acceptance